# Leveraging Plasma Membrane Repair Therapeutics for Treating Neurodegenerative Diseases

**DOI:** 10.3390/cells12121660

**Published:** 2023-06-18

**Authors:** Hannah R. Bulgart, Isabella Goncalves, Noah Weisleder

**Affiliations:** Department of Physiology & Cell Biology, Dorothy M. Davis Heart and Lung Research Institute, College of Medicine, The Ohio State University Wexner Medical Center, Columbus, OH 43210, USA; hannah.bulgart@osumc.edu (H.R.B.); goncalves.25@osu.edu (I.G.)

**Keywords:** membrane repair, neurodegeneration, Alzheimer’s Disease, Parkinson’s Disease

## Abstract

Plasma membrane repair is an essential cellular mechanism that reseals membrane disruptions after a variety of insults, and compromised repair capacity can contribute to the progression of many diseases. Neurodegenerative diseases are marked by membrane damage from many sources, reduced membrane integrity, elevated intracellular calcium concentrations, enhanced reactive oxygen species production, mitochondrial dysfunction, and widespread neuronal death. While the toxic intracellular effects of these changes in cellular physiology have been defined, the specific mechanism of neuronal death in certain neurodegenerative diseases remains unclear. An abundance of recent evidence indicates that neuronal membrane damage and pore formation in the membrane are key contributors to neurodegenerative disease pathogenesis. In this review, we have outlined evidence supporting the hypothesis that membrane damage is a contributor to neurodegenerative diseases and that therapeutically enhancing membrane repair can potentially combat neuronal death.

## 1. Introduction

Cell membrane repair is a cellular process that is essential for cell survival as plasma membranes are constantly disrupted by insults such as mechanical stress, reactive oxygen species (ROS), immune cell infiltration, and pore-forming toxins [1,2]. For cells to remain viable following disruption of the barrier function of the plasma membrane, there must be a process to restore the integrity of the plasma membrane following a membrane injury. Cells which lack effective membrane repair mechanisms can succumb to toxic intracellular alterations that lead to eventual cell death. Without sufficient membrane repair, the integrity of the membrane is compromised which leads to increased membrane permeability [3] and dyshomeostasis of ion concentrations [4]. Fluctuations in ion concentrations can contribute to mitochondrial dysfunction by increased ROS production, which can further damage the plasma membrane and contribute to cell death [5], illustrating why it is essential for cells to maintain effective membrane repair capacity. A defect in the plasma membrane repair mechanism has been known to contribute to the progression of several pathologies, including muscular dystrophies [6,7], kidney injury [8,9], and myocardial injury [10,11,12]. Conversely, enhancing membrane repair capacity has been shown to be beneficial and/or protective in some of these disease states [9,13,14,15,16,17,18].

There are several highly conserved mechanisms that contribute to membrane repair including endocytosis, exocytosis, ectocytosis and patching. Each mechanism functions with some degree of specificity for particular sources of membrane damage and in a cell-type-dependent context. It has been hypothesized that endocytosis is involved in the repair of large membrane disruptions to internalize the damaged section of the membrane to restore membrane integrity, mediated by Endosomal Complexes Required for Transport (ESCRT) machinery [19,20,21] and Rab-5 and -11 [22,23]. Endocytosis has also been described as a response following pore-forming toxin-mediated damage [23,24,25]. For smaller membrane lesions, the damaged membrane can be exported from the cell in a vesicle by budding or ectocytosis [26]. Additionally, membrane repair patch formation is initiated following membrane disruption when extracellular calcium enters the cell through the disruption and initiates intracellular vesicles to traffic to the injury site and undergoes vesicle–vesicle fusion to form a repair patch at the injury site to restore the barrier function of the plasma membrane [27]. Patching is dependent on the exocytotic activity of vesicle trafficking to the membrane and the function of membrane repair proteins associated with the vesicles and/or the plasma membrane. Several proteins have been linked to membrane repair responses, including tripartite motif containing 72/mitsugumin 53 (TRIM72/MG53 [28,29], Annexins A1-A6 [30,31,32,33,34], dysferlin [12,33,35,36] SNAREs [37,38], and synaptotagmin [39,40].

Cell membrane repair has been most extensively studied in skeletal muscle [30] and there have been reports of membrane repair in many other tissue types, including cardiac muscle [41], lung [14,42], and gastrointestinal tract [43]. Membrane repair is relatively understudied in the nervous system as only some recent work has shown that membrane repair is active in neuronal cell types [44] (see Table 1). The presence of a plasma membrane repair response in neurons might be expected, given that these are large cells with limited regenerative capacity. Without a robust mechanism to repair damage to the neuronal cell membrane, neurons would be more susceptible to a variety of insults, such as disruption of ion homeostasis critical for cell signaling, enhanced production of toxic ROS, and mitochondrial dysfunction. It is plausible that a defect in cell membrane repair could contribute to neuronal cell death during the development of neurodegenerative diseases and brain atrophy. This review will focus on the currently available evidence supporting the hypothesis that alterations to membrane integrity are a significant contributor to neurodegeneration, and that leveraging membrane repair as a therapeutic intervention could be beneficial in diseases involving neuronal cell death.

## 2. Neurodegenerative Diseases with Membrane Damage Implications

Neurodegenerative diseases, such as Alzheimer’s Disease (AD) and Parkinson’s Disease (PD), are conditions that involve extensive neuronal loss contributing to memory, behavioral, and motor function deficits. In 2022, it is estimated that about 50 million and 10 million individuals suffer from AD [51] and PD [52], respectively, worldwide. The number of neurodegenerative disease patients is projected to rapidly increase in the near future, with AD incidence growing by approximately 40% [53] and PD by approximately 30% by 2030 [54]. This continued rise of neurodegenerative disease cases poses a major socioeconomic burden on society as it costs hundreds of billions of dollars annually in the United States to care for patients with neurodegenerative diseases [53,54]. Understanding the cellular mechanisms that contribute to disease onset and progression will provide therapeutic targets to reduce disease severity and its socioeconomic impact.

AD is the most common and most severe form of dementia as it leads to short- and long-term memory loss, behavioral alterations (e.g., aggression, depression, etc.), brain atrophy, and eventual death of the patient [53,55,56,57]. As with other neurodegenerative diseases, there is extensive neuronal death in AD, specifically in the hippocampus and neocortex. Two forms of AD exist; the age of onset and genetic factors distinguish them from each other. Familial AD is genetically inherited and rare (~1% of cases), usually affecting patients before the age of 65 years. Familial AD involves specific gene mutations in the APP, PSEN1 and PSEN2 genes. Non-familial AD is the most common form and affects patients older than 65 years of age. Both forms of AD are characterized by the aggregation of two proteins: amyloid beta (Aβ) and neurofibrillary tangles (NFT) composed of hyperphosphorylated tau (p-tau). In a non-AD brain, the full-length Amyloid Precursor Protein (APP) is cleaved by α- then γ-secretases producing non-toxic peptides. Aβ is produced from the amyloidogenic cleavage of the full-length APP peptide by β- then γ-secretases. Aβ can be produced at various sizes between 39 and 42 amino acids in length (Aβ_39–42_), but Aβ_40_ and Aβ_42_ are the most commonly produced isoforms [58]. Aβ_42_ is considered the most pathogenic form as it is insoluble and aggregates to form fibrils in the brain [58]. In the AD brain, there is an imbalance between Aβ production and clearance, which can be caused by mutations in the APP (e.g., KM670/671NL) [59], PSEN1 (e.g., I83T, M84T) [60,61], and/or PSEN2 (e.g., A23A, G34S) [62,63] genes. Aβ is also known to have antimicrobial properties, leading to the thought that Aβ expression could be part of a response to pathogens in the brain and that an excessive Aβ response would lead to plaque formation in AD [64]. NFTs are caused by the hyperphosphorylation of tau, a microtubule-associated protein, which causes tau to separate from the microtubule due to decreased affinity and accumulates intracellularly in the neuron [65]. The risk of tau hyperphosphorylation can be increased by mutations in the MAPT gene (e.g., A152T) [66] which encodes tau. Altered regulation of Aβ and NFTs are contributing factors to neuronal death; however, the specific mechanism by which these protein aggregations lead to cell death remains an area of intensive investigation.

PD, the second most common neurodegenerative disease [52], manifests as muscle tremors and rigidity, slowed movements, and loss of automatic movements. Similar to AD, most PD cases (~90%) are non-familial, but ~10% are familial as they are caused by inherited mutations. Inherited mutations in the SNCA [67], PARK2 [68], PINK1 [69], and PARK7 [70] genes lead to early-onset PD, while most mutations in the LRRK2 [71] gene lead to late-onset PD. The hallmark characteristic of PD is the death or loss of function of dopaminergic neurons, leading to a lack of dopamine, a neurotransmitter involved in movement control. The loss of dopaminergic neurons is thought to be caused by the aggregation of α-synuclein (αS), which is the predominant protein in Lewy bodies [72], clumps of misfolded proteins and molecules in neurons. αS can present as oligomers, protofibrils, and fibril conformations and it is thought the protein conformational state and cellular location of protein accumulation causes specific pathological hallmarks. Three distinct point mutations of αS—A30P, E46K, and A53T—are associated with early onset PD as these mutations enhance αS aggregation in the brain. A30P and A53T are the most consequential mutations as they have the highest increase in fibril formation and aggregation, as compared to the wild-type protein [73]. αs, like Aβ, has also been shown to have antimicrobial properties, which would support an evolutionary purpose for its deposition that could go awry when an elevated response by αs has consequences on nearby tissue. While αS and mutations in other genes are clearly associated with the pathogenesis of PD, the mechanisms by which these changes lead to neuronal death and progression of PD are not clear.

### 2.1. Pathogenic Proteins and the Plasma Membrane

The pathogenic proteins associated with neurodegenerative diseases tend to have a structural and/or functional relationship with the neuronal plasma membrane by which the proteins can cause toxic intracellular conditions that contribute to cell death and the progression of pathology, all of which are summarized in Figure 1. In AD, both hallmark protein accumulations, Aβ and p-tau, are associated with the membrane and it is plausible this association contributes to downstream intracellular alterations by interfering with the barrier function of the membrane. Arispe et al. demonstrated Aβ’s strong binding capacity with the neuronal plasma membrane that could not be removed by several rounds of washing [74]. Furthermore, Aβ specifically binds to lipid rafts enriched with monosialotetrahexosylganglioside (GM1) [75,76]. Aβ binding to the plasma membrane increases Aβ plaque formation [77], which further implicates Aβ–plasma membrane interactions with the progression of AD. Lastly, Julien et al. demonstrated the ability for Aβ to penetrate the plasma membrane in a similar manner as a pore-forming δ-endotoxin, CRY5B. With the use of an innovative *C. elegans* model, the organisms were fed *E. coli* vectors containing Aβ_42_ and CRY5B. It was observed that Aβ_42_ and CRY5B vectors produced similar membrane damage via endosome induction [78]. p-Tau oligomers can also bind to the plasma membrane [79], and also to membrane-bound proteins [80] such as Annexin A2 and A6, two proteins known to be involved in membrane repair [31,33,80,81]. Furthermore, p-tau can directly embed into the membrane and induce damage, just as observed with Aβ [82]. αS, a pathogenic protein involved in PD, also can directly interact with the neuronal plasma membrane as αS binds directly to the plasma membrane to acid phospholipids [83].

### 2.2. Changes in Membrane Permeability, Oxidative Stress and Mitochondrial Dysfunction

Increased membrane permeability is observed in neurodegenerative diseases and such a change would contribute to many potential downstream toxic cellular conditions. Increased membrane permeability indicates the barrier function of the membrane is compromised, producing a dysregulation of the intercellular ion concentrations that are critical for cell signaling and survival. As discussed above, some of the hallmark proteins associated with neurodegenerative diseases can increase membrane permeability, which suggests these proteins may be directly involved in altering the barrier function of the membrane, potentially in a similar fashion as pore-forming toxins. Increased membrane permeability has been observed by a significant increase in lactate dehydrogenase (LDH) release in wild-type human neuroblastoma cells (SH-SY5Y) when Aβ is supplemented [84]. Additionally, Aβ treatment of wild-type neuronal cells quickly results in elevated intracellular calcium levels [85]. In conjunction with increased intracellular calcium levels, membrane conductance increases following Aβ exposure [86]. To directly associate Aβ membrane binding with increased membrane permeability, Evangelisti et al. positively correlated GM1 lipid raft content with the entry of extracellular calcium [76]. Therefore, the greater the GM1 content, the more enriched Aβ is on the neuronal plasma membrane, and the greater the extent of barrier function disruption. In a similar manner that Aβ results in membrane permeability changes, so does tau. Tau oligomer application to wild-type human neuroblastomas induces LDH leakage, indicating a tau-induced membrane integrity defect just like that produced by exposure to Aβ [87]. Application of tau also increased intracellular calcium concentrations, further supporting a decrease in membrane integrity following tau application [88]. In PD, αS is also associated with altering the barrier function of the neuronal plasma membrane. Wild-type and mutant αS recombinant protein application induces increased membrane permeability and decreased membrane integrity marked by dye leakage from lipid vesicles [89,90,91]. Additionally, expression of the A53T αS mutant in human neuroblastomas resulted in increased ion permeability and elevated intracellular calcium concentrations [91]. These findings support the hypothesis that these hallmark proteins of AD and PD modulate the barrier function of the membrane leading to reduced membrane integrity and entry of cytotoxic concentrations of intracellular calcium, leading to downstream effects.

In conjunction with increased membrane permeability and intracellular calcium levels, mitochondrial dysfunction is also observed in neurodegenerative diseases. Calcium and mitochondria have an intertwined relationship as a flux of intracellular calcium through the mitochondria’s outer membrane provides a buffer of intracellular free calcium and the levels of calcium in the mitochondria regulate oxidative phosphorylation. Increased intracellular calcium concentrations cause mitochondria to undergo increased oxidative phosphorylation, which leads to the increased production of ROS and oxidative stress. The mitochondria produce ROS, and, in turn, excess ROS oxidizes mitochondrial components, leading to a cycle of increased dysfunction and oxidative stress levels in the cell. This process is initiated by the plasma membrane being compromised and leading to toxic levels of intracellular calcium, but ROS produce further toxic effects on the cell. One of the many neurotoxic effects of ROS production is lipid peroxidation—the oxidation and degeneration of phospholipids that comprise the lipid bilayer of the plasma membrane. Peroxidation of the plasma membrane compromises the barrier function provided by the lipid bilayer, which poses a great challenge for the cell. Just like the proteins associated with neurodegenerative diseases, increased membrane permeability is observed with lipid peroxidation [92].

Oxidative stress is thought to be a key contributor to neurodegeneration during the progression of AD. High levels of intracellular calcium triggered by Aβ treatment induce the respiratory chain chronically which leads to mitochondrial dysfunction and ROS production [93]. More specifically, Aβ through ROS production activates ASK1, involved in the ER stress signaling pathway, inducing neuronal death via JNK signaling [94,95]. Similarly, tau activates the JNK signaling cascade resulting in apoptosis in an AD *Drosophila* model [96] and JNK activation is observed in AD brain tissue [97]. In a pathogenic cycle, Aβ induces ROS production and mitochondrial dysfunction, which, in turn, promotes Aβ and tau production and accumulation, which then feeds back to further ROS and mitochondrial dysfunction [94]. Just like AD, PD is marked by significant oxidative stress. αS oligomer application to induced pluripotent stem cells (iPSCs) triggered significant ROS production and lipid peroxidation [98]. Previous studies have demonstrated that αS interacts with metal ions independent of other ROS pathways to produce ROS, oxidative stress, and apoptosis in PD neurons. αS binds to metal ions to produce ROS, but when metal chelators are applied, ROS production is blocked [98]. Lastly, ROS enhances αS accumulation in the same manner that ROS enhances Aβ and tau aggregation [99].

### 2.3. Membrane Damage and Pore Formation by Proteins Associated with Neurodegeneration

The available literature describes several indicators of membrane damage—ion dyshomeostasis, ROS production, mitochondrial dysfunction, and lipid peroxidation—in neurodegenerative diseases. Additional data support that Aβ and αS can insert into, and form an unregulated pore in, the neuronal plasma membrane. These pores allow ions to freely flow into and out of these cells, leading to toxic ionic concentrations, altering neuronal excitability, and eventual neuronal death.

The membrane and membrane-associated intracellular alterations observed following Aβ treatment are explained by the so-called “Channel Hypothesis”, pioneered by Nelson Arispe, which moves beyond the previous “Amyloid Cascade Hypothesis” to link Aβ’s toxicity directly to damaging the neuronal plasma membrane by creating a channel or pore. The Channel Hypothesis posits that once Aβ is produced and released in the extracellular space, the peptide has three possible fates—degradation, accumulation, or insertion into the plasma membrane [100]. The hypothesized Aβ peptide insertion would explain the membrane and intracellular alterations explained previously. The Channel Hypothesis is supported by planar lipid bilayer experiments where the Aβ peptide creates a pore and allows the flow of calcium ions through the lipid bilayer, indicating Aβ is sufficient to produce a membrane pore [74]. Additionally, the hypothesis was tested by measuring membrane conductance via the patch clamp technique and Aβ was shown to form ion channels, but this is blocked by the application of Zn^2+^, tromethamine, and aluminum ions [74,101,102]. Importantly, channel formation is positively correlated with increased Aβ aggregation and neurotoxicity [103], which means that factors which reduce neurotoxicity, such as Congo red treatment [104], also reduce channel formation; this observation further supports the Channel Hypothesis’ relevance to AD progression.

This hypothesis is important for the pathogenesis of AD because if neurons are unable to effectively compensate for Aβ-induced membrane damage through membrane repair responses, neurons will succumb to the resulting neuronal death. Additionally, it is plausible that by enhancing neuronal cell membrane repair, Aβ accumulation on/in the membrane would be reduced, resulting in a concurrent decrease in Aβ toxic effects. Neuronal cell membrane damage via Aβ is detrimental to neurons as it disturbs ion concentrations, specifically observed with calcium. The elevated concentration of intracellular calcium interferes with action potential firing, cell signaling, and leads to apoptotic pathways. A sufficient membrane repair response is required to repair the damage caused by Aβ to the neuronal membrane to restore the barrier function of the membrane and inhibit extracellular calcium ions from continuing to flow into neurons.

αS has also been shown to form an unregulated pore in the neuronal plasma membrane similar to Aβ [105]. Interestingly, pore formation in the plasma membrane, as determined by a significant increase in membrane conductance levels, is only observed with oligomer structures and not with monomers or fibrils [106]. Multiple studies have identified the size-dependent permeabilization capacity of αS and mutant oligomers, which has been argued as evidence for pore formation. It has been observed that only small molecules such as calcium and dopamine can enter the neurons in the presence of αS and mutant oligomers, but not larger molecules such as FITC-dextran [107]. In a molecular dynamic simulation, Tsigelny et al. determined that αS embeds into the membrane in a pore-like structure consisting of an αS octamer in a hollow ring. In addition to molecular dynamic simulations, their group also observed punctae in wild-type α- and A53T-expressed cells, and compromised membrane permeability as indicated by increased intracellular calcium levels [108]. Lastly, membrane conductance levels observed with αS pore are very similar to those produced by exposure to pore-forming toxins [109]. All this evidence supports that αS can penetrate the plasma membrane, form a pore in the neuronal plasma membrane, and induce neurotoxic conditions leading to the death of dopaminergic neurons and the progression of neurodegeneration.

### 2.4. Membrane Repair Proteins Involved in Neurodegeneration

In addition to physiology-associated findings connecting neurodegeneration and membrane repair, there are also membrane repair-associated indicators in AD and PD. Apolipoprotein E (APOE) gene variant ε4 is a very well-known risk factor for late-onset AD [110]. Interestingly, APOE has been thought to be involved in membrane repair as it is associated with the ESCRT mechanism. It has been demonstrated that APOE facilitates cellular internalization of Aβ and packaging into lysosomes [111,112,113], proving it could be involved in the ESCRT pathway. Additionally, some of the annexin proteins and their variants have been suggested as potential markers of AD. An inactive or cleaved version of annexin A1 protein has been identified in the neocortex of AD brains and positively correlated with Aβ [114,115]. In PD, missense mutations in the ANXA A1 gene are rare but identified in certain patients [116]. Elevated levels of annexin A5 in plasma were identified in AD patients and dementia with Lewy bodies as compared to healthy controls [117], indicating potential overexpression of the ANXA A5 gene. However, there is reduced annexin A5 in the cerebrospinal fluid of PD patients [118]. Lastly, variants of the dysferlin gene have been associated with AD susceptibility [119]. In addition to dysferlin variants, it has been demonstrated that the dysferlin protein accumulates in the AD brain and co-localizes with Aβ [120]. The alterations in membrane repair proteins in AD and PD expand the support for the involvement of membrane repair in the pathogenesis of neurodegenerative diseases.

## 3. Enhancing Cell Membrane Repair as a Therapeutic Approach

Several lines of evidence support that hallmark proteins of AD and PD interfere with neuronal homeostasis. Accumulations of these proteins have an intimate relationship with the neuronal plasma membrane where they can induce damage to the membrane and produce toxic intracellular effects. Since the cell compensates for such membrane disruptions using membrane repair responses it is possible that increasing membrane repair could reduce neurotoxicity and neuronal cell death in neurodegenerative diseases (Figure 2). Thus, targeting membrane repair as a therapeutic invention has the potential to reduce the neurotoxic events that follow membrane damage in neurodegenerative disease. While changes in membrane repair capacity have not been directly studied in the context of AD and PD, membrane repair therapeutics have been preliminarily studied in vitro and in vivo in models of these diseases, with mixed results. Recombinant MG53, in combination with mesenchymal stem cells, treated APP/PS1 mice showed improved memory, reduced Aβ deposition and p-tau, and reduced oxidative stress [121]. Recombinant annexin A1 treated 5X FAD and P301L mice showed increased blood–brain barrier integrity, reduced Aβ_40_, improved memory, and reduced p-tau [122]. Poloxamer 188 (P188), a membrane sealant, has been shown to promote neuronal membrane repair, increase cell survival, and reduce toxicity in vitro with neuronal cell types treated with Aβ and αS [123]. However, high doses of P188 were shown to increase Aβ deposition in vivo in 5X FAD mice, most likely by enhancing γ-secretase cleavage of APP [124]. More comprehensive studies will be needed to determine the therapeutic potential of these agents in improving AD/PD pathology. Several approaches have been used to enhance membrane repair in multiple tissue types, some of which we will highlight in the following section.

In general, there are two approaches that have been used in the past to therapeutically enhance membrane repair—application of recombinant membrane repair proteins and application of synthetic molecules (summarized in Table 2). One of the most commonly used therapeutics is recombinant TRIM72, also known as MG53. TRIM72 is a membrane repair protein highly expressed in striated muscle that traffics to the injury site to repair the membrane lesion as part of the formation of a membrane repair patch [29]. The application of recombinant TRIM72 protein (also known as rhMG53) enhances membrane repair kinetics in striated muscle [16,17] and tissue with no native TRIM72 expression [14]. Recombinant TRIM72 enhances membrane repair by accumulating at the injury site where it binds to the inner leaflet of the plasma membrane to create a seal [16]. Recombinant TRIM72 has been tested in many tissues including in the peripheral nervous system where application of the protein before nerve crush injury can increase sciatic nerve regeneration after injury [44]. Additionally, recombinant annexin proteins have been utilized to enhance membrane repair in vitro and in vivo. The Annexin family of proteins are calcium-sensitive proteins, which, like TRIM72, accumulate in membrane repair patches at the injury site of membrane damage [125]. Annexins have a high binding affinity to many membrane components, including phosphatidylserine, cholesterol and the protein dysferlin, which allows annexins to effectively participate in repair [33,125]. Recombinant annexin A6 has been used as a therapeutic utilized to enhance membrane repair and myofiber survival following cardiotoxin injury in wild-type and muscular dystrophy mouse models in vivo [125]. Additionally, recombinant annexin A6 enhances membrane repair in dystrophic iPSC-derived cardiomyocytes in vitro [126]. In addition to recombinant TRIM72 and annexin A6, other recombinant membrane repair proteins have been utilized to enhance membrane repair, which are outlined in Table 2.

Tri-block copolymers, or poloxamers, are non-ionic synthetic molecules that self-assemble when dissolved in a physiological buffer and form a hydrophobic head which can interact with the plasma membrane, and hydrophilic chains that interact with an aqueous solution. There are a wide variety of different poloxamer molecules that are distinguished by molecular weight and the ratio of hydrophobic/hydrophilic properties. Poloxamers with long hydrophobic chains can deeply embed into the membrane and actually increase membrane permeability, but poloxamers with long hydrophilic chains have a greater affinity for the outer leaflet of the membrane where they can effectively increase the integrity of the plasma membrane [127]. Poloxamers are effective in restoring the integrity of the membrane with membrane damage as they act as a sealant to membrane wounds in addition to native membrane repair [128]. P188 is the most common synthetic therapeutic used to enhance membrane integrity in vitro and in vivo. P188 has long hydrophilic chains indicating a low affinity for the inner leaflet of the plasma membrane. Many research groups show this in skeletal muscle [129] and cardiac muscle [130]. P188 has also been shown to increase membrane repair in neuronal cell types in vitro after laser ablation injury [44]. A recent study by Kwiatkowski et al. demonstrated P188 and several of the related poloxamers are highly efficacious to enhance membrane repair in vitro with HEK293 cells and in ex vivo skeletal muscle isolated from a muscular dystrophy mouse model (*mdx*) [131]. The basic science research on P188 led to it being tested as a therapeutic for DMD patients in a clinical trial (identifier NCT03558958)

The nervous system as a whole is understudied when it comes to membrane repair. Very little work has been done to assess the membrane repair capacity of the cells of the nervous system, and what proteins play a critical role in the mechanism. Membrane repair is critical to neuronal cells in the central nervous system particularly since most are terminally differentiated and there is little to no regeneration available to compensate for the loss of neurons. Paleo et al. conducted one of the first direct investigations of membrane repair capacity in neuronal cell types and observed the patch formation mechanism when injuring mouse neuroblastomas, which had never been shown before. Additionally, membrane repair therapeutics recombinant TRIM72 and P188 were shown to enhance membrane repair in vitro and in vivo [44]. These studies support that therapeutically enhancing membrane repair in neurodegenerative diseases could have the capacity to reduce neuronal death, brain atrophy, and continual cognitive decline. This would allow for the plasma membrane damage induced by Aβ, p-tau or αS to be repaired effectively and minimize the death of neurons. Effectively enhancing membrane repair would increase the integrity of the membrane, which would normalize intracellular ion concentrations at physiologic levels, reduce ROS production and prevent mitochondrial dysfunction. We believe therapeutically targeting membrane repair capacity in neurodegenerative diseases represents a promising therapeutic approach to curbing AD and PD.
cells-12-01660-t002_Table 2Table 2Commonly used membrane repair therapeutics.MoleculeTarget Tissue(s)Effect on Membrane RepairReferencesRecombinant TRIM72/MG53Heart, skeletal muscle, kidney, liver, peripheral nervous systemIncreases membrane repair capacity; accumulates at the injury siteWeisleder et al. (2012) [16], Gushchina et al. (2017) [17], Paleo et al. (2020) [44]Recombinant Annexin A6Skeletal muscleEnhances membrane repair, protects against skeletal muscle damageDemonbreun et al. (2019) [125]Recombinant Annexin A5MyotubesRescues membrane repair from annexin A5 knockdownCarmeille et al. (2016) [30]Recombinant Annexin A1HeartReduced ischemia-reperfusion damageD’Amico et al. (2000) [132]Recombinant Annexin A2BrainEnhances blood brain barrier integrityCheng et al. (2021) [133]P188Skeletal muscle, cardiac muscle, lung, brainIncreases membrane resealingMoloughney and Weisleder (2012) [128], Kwiatkowski et al. (2020) [131], Spurney et al. (2011) [130], Tang et al. (2021) [134], Gu et al. (2013) [135]

## 4. Conclusions

Plasma membrane repair is an essential physiological process required to compensate for membrane injury to allow for cell survival. Defects in efficient cell membrane repair can contribute to the progression of certain pathologies, including inherited myopathies, inflammatory myopathies, lung injury and others, due to increased cell death events. Cell membrane repair is an evolutionarily conserved mechanism; however, it has only been heavily studied in a limited number of tissue types. Plasma membrane repair has not been extensively studied in the nervous system, and no studies have addressed it in the context of neurodegenerative diseases. In this review, we described various intracellular effects produced by a decrease in membrane integrity and membrane repair protein alterations that could contribute to the progression of neurodegenerative diseases. The available evidence demonstrates the tight interaction of Aβ, p-tau, and αS with the plasma membrane. This intimate relationship between hallmark proteins of neurodegenerative disease and the plasma membrane results in increased membrane permeability, which leads to toxic concentrations of intracellular calcium, mitochondrial dysfunction, and increased ROS production that can peroxidize the plasma membrane. Additionally, pore formation in the plasma membrane has gained support as a mechanism contributing to neurodegenerative diseases, which could explain many of the intracellular alterations observed in these diseases. We hypothesize that enhancing membrane repair via therapeutic molecules could improve the integrity of the plasma membrane, normalize intracellular ion concentrations, reduce neuronal cell death and treat neurodegenerative diseases.

## Figures and Tables

**Figure 1 cells-12-01660-f001:**
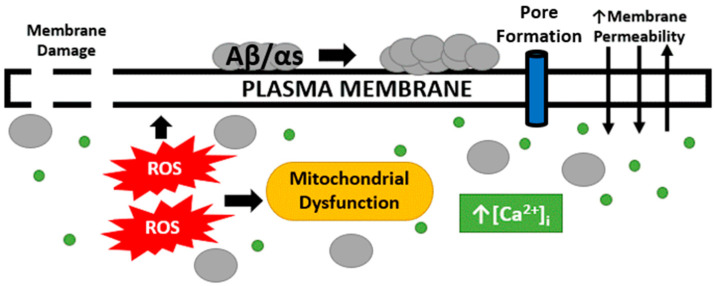
Plasma membrane and intracellular alterations following Aβ and αS exposure. Following neurodegenerative disease hallmark proteins binding to the neuronal plasma membrane, AD and PD neurons display enhanced membrane permeability, marked by toxic concentrations of intracellular calcium, which then contributes to increased ROS production and mitochondrial dysfunction. AD and PD hallmark proteins have also been shown to penetrate and damage the plasma membrane. Additionally, it has been hypothesized to form a pore in the membrane as an additional form of damage.

**Figure 2 cells-12-01660-f002:**
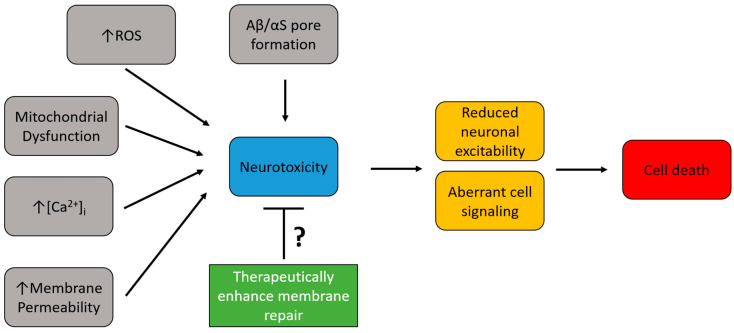
Therapeutic potential of enhancing neuronal plasma membrane repair in neurodegenerative diseases. Neurodegenerative diseases are marked by several toxic intracellular alterations (gray boxes) that lead to neurotoxic conditions (blue box). This neurotoxicity has detrimental effects on neuronal functions (yellow boxes), which ultimately lead to cell death (red box) by apoptosis. Enhancing membrane repair (green box) has the potential to reduce the observed neurotoxicity by increasing plasma membrane integrity.

**Table 1 cells-12-01660-t001:** Conserved membrane repair responses in multiple tissue types.

Organ/Tissue	Type of Damage	Cell Type	References
Skeletal muscle	Aperiodic, eccentric contractions, saponin, chronic damage (e.g., muscular dystrophy)	Myocytes, myotubes	Cooper and McNeil (2015) [45], Carmeille et al. (2016) [30], Defour et al. (2014) [36]
Cardiac muscle	Ischemia/reperfusion injury	Cardiomyocytes	Houang et al. (2019) [11], Han et al. (2007) [12]
Skin	Aperiodic	Epidermal cells, fibroblasts, etc.	McNeil and Ito (1990) [46], Reddy et al. (2001) [47]
Gastrointestinal tract	Cyclic	Epithelial cells, smooth muscle cells	McNeil and Ito (1989) [43], Ammendolia et al. (2021) [48]
		data	
Respiratory	Stretch, overventilation, saponin	Epithelial cells, endothelial cells, smooth muscle cells	Ammendolia et al. (2021) [48], Cong et al. (2017) [49], Cong et al. (2020) [13]
Peripheral Nervous System	Crush injury, nerve transection	Sciatic nerve, Schwann cells	Paleo et al. (2020) [44], Rigonia and Negro (2020) [50]

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
