# Peer review of "Leveraging Plasma Membrane Repair Therapeutics for Treating Neurodegenerative Diseases"

_cells, 2023, doi:10.3390/cells12121660_

Round 1
Reviewer 1 Report
The authors provide an overview of the importance of membrane damage and repair mechanisms in neurodegenerative diseases and suggest that targeting membrane repair could have therapeutic implications. Overall it’s a good review.
Here I have a few comments:
- The authors primarily focus on Alzheimer's Disease (AD) and Parkinson's Disease (PD) as examples of neurodegenerative diseases in the paper. It would be beneficial if the authors could also discuss other neurodegenerative diseases, such as Huntington's disease, to provide a more comprehensive understanding of the potential implications of membrane damage and repair mechanisms across different conditions.
- While the authors summarize and discuss the efforts made to improve cell membrane repair in the paper, it would be valuable to provide additional comments on the therapeutic effects achieved thus far in neurodegenerative diseases. Specifically, discussing whether enhancing membrane repair has shown promising results in neurodegeneration and improving clinical outcomes. Furthermore, it would be interesting to highlight any limitations or areas that have not been extensively explored in this context.
- Targeting/eliminating amyloid-beta (Aβ) is a common therapeutic approach for Alzheimer's Disease (AD) treatment. It would be helpful to elaborate on the connection between membrane repair and Aβ elimination. Specifically, discussing how enhancing membrane repair mechanisms could potentially inhibit of Aβ accumulation, and how this could impact the progression and treatment of AD.
Minor editing of English language required
Author Response
We appreciate the insight provided by the reviewers of our initial submission. To address the concerns raised we have revised components of the manuscript. Changes to the manuscript are indicated by tracked changes and are detailed below:
- “Line 362: Is it possible for diseases like AD or PD to occur when there is a deficiency in the repair of neuronal cell membranes?” (Editor); “If membrane repair plays a role in neurodegenerative diseases, it would be predicted that genes involved in this process would harbor risk alleles. Is there any support for this? ApoE has been claimed to be involved in membrane repair due to its role in lipid transport, and mutations CHMP2A, a gene involved in the ESCORT complex, can cause ALS/FTD. Other examples would strengthen the argument.” (Rev 2)
We appreciate the suggestions on these topics. To address these concerns, we have addressed these two questions by adding a new subsection titled “Membrane repair proteins involved in neurodegeneration” which details the known membrane repair transcript/protein alterations in the available literature. Addressing these concerns on membrane repair protein/transcript alterations further strengthens our argument of the implication of membrane repair in neurodegenerative diseases and has improved the impact of this review.
- “While the authors summarize and discuss the efforts made to improve membrane repair in the paper, it would be valuable to provide additional comments on the therapeutic effects achieved thus far in neurodegenerative diseases. Specifically, discussing whether enhancing membrane repair has shown promising results in neurodegeneration and improving clinical outcomes. Furthermore, it would be interesting to highlight any limitations or areas that have not been extensively explored in this context.” (Rev 1)
To address this concern, we added more information to the first paragraph of the “Enhancing cell membrane repair as a therapeutic approach section” to highlight membrane repair therapeutics that have been tested in neurodegenerative models. While there has been very little work done in this area, we did detail any studies that make use of recombinant MG53/TRIM72, recombinant annexin A1, and Poloxamer 188 in AD and PD cell and animal models.
- “Targeting/eliminating Aβ is a common therapeutic approach for AD treatment. I would be helpful to elaborate on the connection between membrane repair and Aβ elimination. Specifically, discussing how enhancing membrane repair mechanisms could potentially inhibit Aβ accumulation, and how this could impact the progression and treatment of AD”. (Rev 1)
To address this concern, we added the following sentence to subsection 2.3: “Additionally, it is plausible that by enhancing neuronal cell membrane repair, Aβ accumulation on/in the membrane would be reduced, along with related Aβ toxicities.” There is no direct evidence to suggest that enhancing cell membrane repair would eliminate Aβ, but it is plausible that increasing cell membrane repair could reduce Aβ in the membrane.
- “The authors primarily focus on AD and PD as examples of neurodegenerative diseases in the paper. It would be beneficial if the authors could also discuss other neurodegenerative diseases, such as Huntington’s Disease, to provide a more comprehensive understanding of the potential implications of membrane damage and repair mechanisms across different conditions.” (Rev 1)
We appreciate this comment by the reviewer. While AD and PD are certainly not the only neurodegenerative diseases, we chose to focus on these two diseases as a case study for our arguments as their pathology is very similar and as they are the two most common neurodegenerative diseases. A comprehensive assessment of several more neurodegenerative disease would go beyond the scope of this review that is focused on these two diseases.
- “Both Aβ and αs have been claimed to have antimicrobial activity. Most antimicrobial peptides are amphipathic and disrupt bacterial membranes, suggesting a functional reason why Aβ and αs might damage plasma membranes.” (Rev 2)
We appreciate this comment by the reviewer. We have added text commenting on the antimicrobial properties of these peptides during the introduction of each of these two diseases.
- “Line 347-348: since this experimental result is what authors want to emphasize the most in this paper, please provide a more detailed explanation”. (Editor)
We have addressed this concern by adding additional text to the conclusion paragraph that specifically details the experimental results that support our proposed conclusions.
Reviewer 2 Report
This review makes a good case for a possible role of membrane damage/repair in neurodegenerative diseases. The references are appropriate and the interpretation of the studies described is reasonable. There is one major mistake (see below), and there are a couple of points that, if considered, would enhance the review. Specifically:
1) If membrane repair plays a role in human neurodegenerative diseases, it would be predicted that genes involved in this process would harbor risk alleles. Is there any support for this? ApoE has been claimed to be involved in membrane repair due to its role in lipid transport, and mutations CHMP2A, a gene involved in the ESCORT complex, can cause ALS/FTD. Other examples would strengthen the argument.
2) Both Abeta (e.g., https://pubmed.ncbi.nlm.nih.gov/27225182/) and alpha-synuclein (e.g., https://pubmed.ncbi.nlm.nih.gov/27520375/) have been claimed to have antimicrobial activity. Most antimicrobial peptides are amphipathic and disrupt bacterial membranes, suggesting a functional reason why Abeta and alpha synuclein might damage plasma membranes. Making this point would also strengthen the contention of the review.
Major correction (lines 96-98):
Familial AD involves mutations in APP or PSEN1 & 2 (not non-familial AD!). MAPT mutations cause frontotemporal dementia, not Alzheimer's. There are risk alleles in MAPT for both Alzheimer's and Parkinson's.
Author Response

(The authors gave the same response as above.)
